# The incidence of admission ionised hypocalcaemia in paediatric major trauma— A systematic review and meta-analysis

Owen Hibberd[1,2]*, James Price[1,3], Stephen H. Thomas[2,4], Tim Harris[2], Edward B. G. Barnard[1,3,5]

1 Emergency and Urgent Care Research in Cambridge (EURECA), PACE Section, Department of Medicine, Cambridge University, Cambridge, United Kingdom, 2 Blizard Institute, Queen Mary University of London, London, United Kingdom, 3 Department of Research, Audit, Innovation, & Development (RAID), East Anglian Air Ambulance, Norwich, United Kingdom, 4 Department of Emergency Medicine, Beth Israel Deaconess Medical Center & Harvard Medical School, Boston, MA, United States of America, 5 Academic Department of Military Emergency Medicine, Royal Centre for Defence Medicine (Research and Clinical Innovation), Birmingham, United Kingdom

* oh296@cam.ac.uk

**Data Availability Statement:** All relevant data are within the paper and its supporting information files.

## Abstract

### Objectives

In adult major trauma patients admission hypocalcaemia occurs in approximately half of cases and is associated with increased mortality. However, data amongst paediatric patients are limited. The objectives of this review were to determine the incidence of admission ionised hypocalcaemia in paediatric major trauma patients and to explore whether hypocalcaemia is associated with adverse outcomes.

### Methods

A systematic review was conducted following PRISMA guidelines. All studies including major trauma patients <18 years old, with an ionised calcium concentration obtained in the Emergency Department (ED) prior to the receipt of blood products in the ED were included. The primary outcome was incidence of ionised hypocalcaemia. Random-effects Sidik-Jonkman modelling was executed for meta-analysis of mortality and pH difference between hypo- and normocalcaemia, Odds ratio (OR) was the reporting metric for mortality. The reporting metric for the continuous variable of pH difference was Glass' D (a standardized difference). Results are reported with 95% confidence intervals (CIs) and significance was defined as $p < 0.05$.

### Results

Three retrospective cohort studies were included. Admission ionised hypocalcaemia definitions ranged from <1.00 mmol/l to <1.16 mmol/l with an overall incidence of 112/710 (15.8%). For mortality, modelling with low heterogeneity ($I^2$ 39%, Cochrane's Q $p = 0.294$) identified a non-significant ($p = 0.122$) estimate of hypocalcaemia increasing mortality (pooled OR 2.26, 95% CI 0.80–6.39). For the pH difference, meta-analysis supported

**Funding:** The author(s) received no specific funding for this work.

**Competing interests:** The authors have declared that no competing interests exist.

generation of a pooled effect estimate ($I^2$ 57%, Cochrane's Q $p = 0.100$). The effect estimate of the mean pH difference was not significantly different from null ($p = 0.657$), with the estimated pH slightly lower in hypocalcaemia (Glass D standardized mean difference -0.08, 95% CI -0.43 to 0.27).

## Conclusion

Admission ionised hypocalcaemia was present in at least one in six paediatric major trauma patients. Ionised hypocalcaemia was not identified to have a statistically significant association with mortality or pH difference.

## Introduction

Major trauma is a leading cause of death in children [1, 2]. Haemorrhage is a key aetiology of potentially survivable death following trauma, and it is well recognised that the 'lethal triad' of coagulopathy, hypothermia, and acidosis can further potentiate haemorrhage [3, 4]. Recently it has been recognised that ionised hypocalcaemia (iHypoCa) also contributes to this 'lethal triad', and as such this may now be considered the 'diamond of death' [5–7]. An iHypoCa contributes to coagulopathy and cardiovascular decompensation due to its role in clot formation, vascular tone, and cardiac contractility [6–8]. Consequently, current guidelines recommend maintaining normal calcium levels in the bleeding trauma patient [9].

There is good agreement between arterial and venous calcium measurement on blood gases, which are often obtained shortly after hospital arrival in severely injured paediatric patients [10]. Blood gases report ionised calcium (iCa), which is the free-form and physiologically active calcium in the blood [11]. Transfusion of citrated blood products leads to iHypoCa due to calcium chelation with citrate [12]. However, in trauma, early iHypoCa may occur in the absence of exogenous citrate and the aetiology is complex, with multiple potential mechanisms [12, 13]. Pathophysiological mechanisms include dilution by crystalloid fluid administration, cellular calcium shifts secondary to ischaemia and reperfusion injury, calcium-lactate binding, and impaired calcium homeostasis [5–7]. In adult major trauma patients iHypoCa occurs in approximately half of cases and is associated with coagulopathy, increased blood transfusion requirements, and increased mortality [14–19] However, in paediatric major trauma patients data are limited. In critically ill paediatric patients on the Paediatric Intensive Care Unit (PICU) iHypoCa has been observed to be independently associated with more severe organ dysfunction and more frequent amongst non-survivors compared to survivors [20, 21]. In trauma, children may be more sensitive to iHypoCa due to different injury kinematics and developmentally dynamic physiological responses to injury [22, 23]. Therefore, it may not be appropriate to extrapolate results from studies involving adult major trauma patients to the paediatric major trauma patient.

The objective of this systematic review and meta-analysis was to determine the incidence of admission iHypoCa in paediatric major trauma patients and explore whether this is associated with adverse clinical outcomes when compared to normocalcaemia (iNormoCa).

## Materials and methods

This study adheres to the Preferred Reporting Items for Systematic Reviews and Meta-Analyses (PRISMA) checklist (S1 Checklist) [24].

## Eligibility criteria

This systematic review explores the incidence of iHypoCa in paediatric major trauma patients and whether admission iHypoCa (iCa <1.16 mmol/l), compared to iNormoCa (iCa ≥1.16 mmol/l) is associated with a greater incidence of adverse outcomes. Paediatric (<18 years old) major trauma patients (Injury Severity Score >15, or trauma team activation) with a documented iCa concentration on admission to the emergency department (ED) were included. Studies involving patients who had an iCa concentration obtained after the administration of blood products in the ED were excluded, in keeping with the methodology of a systematic review of adult studies [17]. The full Population, Intervention, Comparison, Outcomes, and Study Design (PICOS) eligibility criteria are available in the previously published protocol [25].

## Information sources and search strategy

A literature search for articles in the English language was completed using MEDLINE on the EBSCO platform, CINAHL on the EBSCO platform, and Embase on the Ovid platform. Articles were searched from database inception to the search date (03/07/2023). The reference lists of all included studies, and the grey literature were also searched. The search strategy can be found in the online supplementary tables of the published protocol [25].

## Selection process

The search strategy was undertaken by a trained librarian from the UK Defence Medical Services Burnett Library (S1 File). Duplications were manually removed and the combined abstracts from the search strategy were then independently screened by two reviewers (OH and JP) to identify studies that met the inclusion criteria. For abstracts meeting inclusion criteria, full texts were retrieved and independently screened against the eligibility criteria by two reviewers and an adjudicating third reviewer (EB).

## Data collection process

A standardised data sheet (Microsoft ® Excel for Mac, Version 16.72, 2023) was used to extract data from included studies to facilitate data synthesis and assessment of quality and risk of bias (S2 File). Extracted data were independently verified by the second reviewer, and any discrepancies were adjudicated by the third reviewer. For incomplete data, or different reporting of central tendency (e.g., mean, and median) the corresponding authors of the articles were contacted directly for additional information.

## Data items

The full protocol, detailing data items extracted, is published and available online [25]. The Microsoft Excel® data sheet containing full extracted data can be found in S2 File.

## Study risk of bias assessment

The risk of bias was assessed for all included studies using the ROBINS-I tool [26]. The risk of publication bias was assessed with funnel plots. The quality of evidence was assessed using the Grading of Recommendations Assessment, Development and Evaluation (GRADE) approach [27]. The quality of evidence for each study was also assessed independently by two authors, with consensus agreement or adjudication by a third author for cases of disagreement.

## Effect measures

The primary outcome of this systematic review was the incidence of admission iHypoCa. Secondary outcomes are the associations with adverse outcomes and physiological abnormalities. The endpoint for adverse outcomes was classified dichotomously as the odds of in-hospital mortality. Treatment requirements (such as need for transfusion) and the LOS (hospital and PICU) were also compared between studies. The end point for physiological abnormalities was the mean difference in pH, this was driven by available data and selected due to universal reporting across studies. Other physiological abnormalities compared between studies were markers of coagulopathy, lactate, and the odds of haemodynamic instability defined as an elevated Shock Index Paediatric Age-Adjusted (SIPA) (0-6yrs >1.22, 7-12yrs >1.00, 13-16yrs >0.90) [28, 29].

Ratio measures (odds ratio) and mean differences are used for measures of effect. Where possible, the estimate of effect was presented along with a confidence interval and a *p*-value.

## Synthesis methods

Data were synthesised as per the PRISMA guidelines [24]. Studies were assessed clinically (PICO) and methodologically (study design, comparability, outcome ascertainment, and risk of bias). A narrative synthesis and summary of effect measures were conducted when heterogeneity was deemed too substantial for meaningful meta-analysis. All analysis and plotting were executed with Stata (Stata, version 18MP, 2024, Stata Corp, College Station TX, USA; www. stata.com). To assess the endpoints, we executed random effects modelling. After the number of studies (N) was determined to be small, we employed a Sidik-Jonkman (SJ) approach recommended for small-N meta-analysis [30]. The SJ approach was chosen due to its superior performance for random effects meta-analysis, especially in small studies [31]. Sensitivity to model selection was assessed by re-executing meta-analysis using a second random effects approach, the DerSimonian-Laird model. A third modelling approach, fixed-effect modelling, was executed solely as an indicator of study heterogeneity (i.e., to determine whether results were markedly different from random effects results). The assessment method used for comparing continuous outcomes (e.g., pH differences) was the Glass D metric standardized by control group. This approach was chosen conservatively, since the standard deviations between iHypoCa and iNormoCa groups differed substantially in one study (Epstein). We also report an effect estimate of the non-standardized mean between pH in iHypoCa- vs. iNormoCa groups in S3 File.

Formal heterogeneity and small-study bias (including publication bias) were evaluated using $I^2$ (with 30–60% representing "potentially moderate" heterogeneity), Galbraith plotting, and funnel plots with trim-and-fill (imputed study) analysis [32]. Sensitivity analyses included omitted-study plotting and cumulative meta-analysis. Meta-regression and prediction intervals were planned in the event the analysis accrued the minimum requirement of ten studies [29].

## Results

### Study selection

A total of 22 abstracts were screened through a database search, reference review, and the grey literature (Fig 1).

Five unique full-text studies were examined, two of which were excluded [33–37]. One study was excluded as it reported total calcium rather than ionised calcium [33]; the other as blood products in the Emergency Department (ED) were received prior to calcium

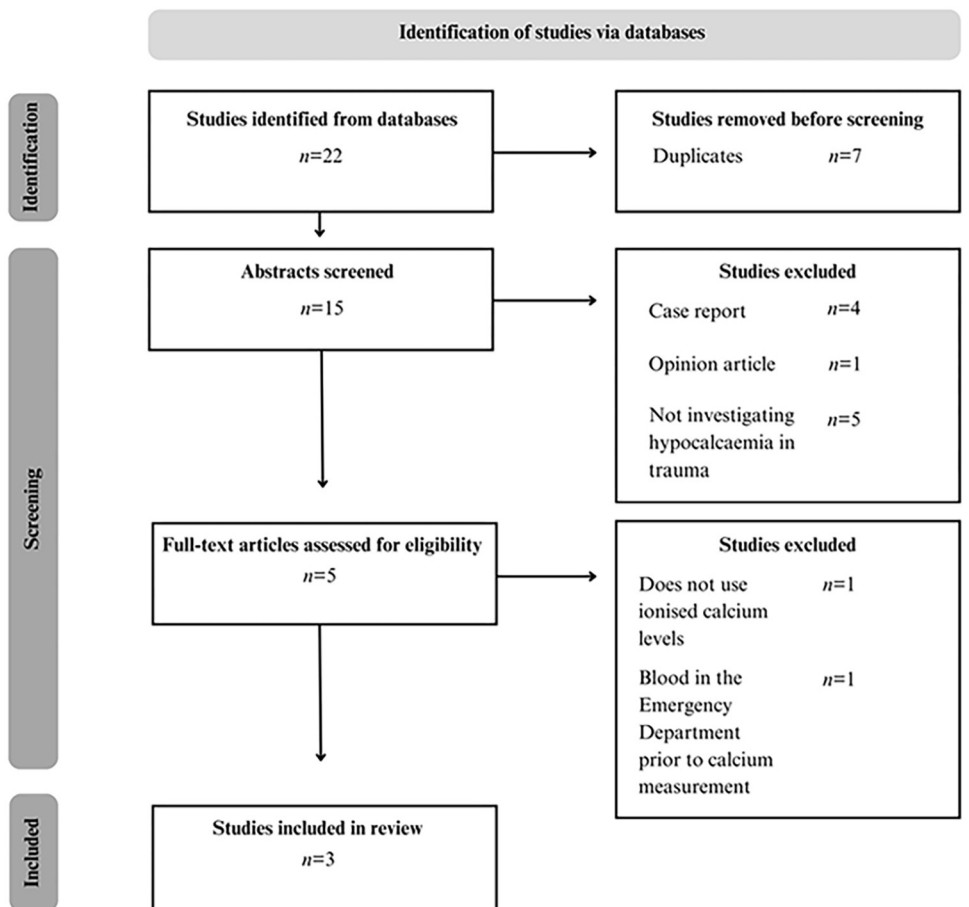

**Fig 1. Preferred reporting items for systematic reviews and meta-analyses flow diagram for paediatric major trauma and associations with admission ionised hypocalcaemia.**

measurement [34]. This resulted in three studies being included in the review [35–37]. All three studies were retrospective cohorts (Table 1) undertaken at a paediatric trauma centre and the quality of evidence was either low or moderate, owing to confounders, risk of bias, and sample size [35–37]. Studies differed in sample size (n = 111–457), inclusion criteria related to trauma team activation, and prevalence of penetrating injury (4%-24%) [35–37].

## Assessment of risk of bias

Overall, although all studies were well conducted, a serious to critical risk of bias was identified (Table 2). Bias was largely due to confounding, with differential determinants being unrecorded, for example, the interval time from injury to calcium measurement and pre-hospital treatments were not reported in two studies [36, 37]. Moreover, the selection of patients who received trauma team activation also risked missing major trauma patients who self-present or those with occult injuries identified on subsequent imaging.

## Incidence of admission ionised hypocalcaemia

Admission iHypoCa was defined differently by each study (Table 3). Definitions ranged from <1.00 mmol/l to <1.16 mmol/l [35–37]. The overall incidence of iHypoCa was 112/710 (15.8%) in the emergency department for paediatric major trauma patients.

**Table 1. Study and patient characteristics for paediatric major trauma and associations with admission ionised hypocalcaemia.**

| Author | Study type and Quality of Evidence (GRADE) | Study setting | | | Summary of Inclusion Criteria | | Summary of Exclusion Criteria | | | Cohort size | Age in years | | | Injury Severity Score | | | Blunt injury |
|---|---|---|---|---|---|---|---|---|---|---|---|---|---|---|---|---|---|
| | | | | | | | | | | | iHypoCa | iNormoCa | p | iHypoCa | iNormoCa | p | |
| Ciaraglia et al. 2023 | Low quality of evidence | Retrospective cohort study | 2016–2021 | Level I Paediatric Trauma Centre | USA | Requiring the highest level of the institution's trauma team activation | <18yrs | Isolated severe head injury | Died within 15 minutes of arrival | Transferred from another facility | n = 142 | Mean 9.56 (SD 4.88) | Mean 6.89 (SD 5.39) | 0.002 | Mean 25.73 (SD 11.8) | Mean 23.77 (SD 11.2) | 0.37 | 108/142 (76%) |
| Epstein et al. 2022 | Moderate quality of evidence | Retrospective cohort study | 2012–2020 | Level I Paediatric Trauma Centre | Israel | Requiring trauma team activation | ISS >15 | <18 years | Receipt of blood products prior to iCa measurement | Transferred from another facility | n = 457 | Mean 13.0 (SD 4.63) | Mean 9.9 (SD 5.2) | 0.003 | Mean 31.83 (SD 12.79) | Mean 27.4 (SD 10.16) | 0.05 | 439/457 (96%) |
| Gimelraikh et al. 2022 | Low quality of evidence | Retrospective cohort study | 2010–2020 | Level I Paediatric Trauma Centre | Israel | Requiring trauma team activation | ISS >15 | Isolated burns | Receipt of blood products prior to iCa measurement | Transferred from another facility | n = 111 | Mean 9 (SD 5) | Mean 10 (SD 5.47) | 0.9 | Mean 27 (SD 11) | Mean 24 (SD 10.9) | 0.39 | 106/111 (96%) |

GRADE—Grading of Recommendations Assessment, Development and Evaluation, iCa–Ionised calcium; iHypoCa–Low ionised calcium concentration–defined differently by each author; iNormoCa–Normal ionised calcium concentration–defined differently by each author; ISS–Injury Severity Score; SD–standard deviation

**Table 2. Risk of bias assessments for each study on paediatric major trauma and associations with admission ionised hypocalcaemia.**

| Author | Bias due to confounding | Bias in the selection of participants for the study | Bias in the classification of interventions | Bias due to deviations from intended interventions | Bias due to missing data | Bias in the measurement of the outcome | Bias in the selection of the reported result | Overall Risk of Bias |
|---|---|---|---|---|---|---|---|---|
| Ciaraglia *et al.* 2023 | Serious risk of bias | Moderate risk of bias | Low risk of bias | Low risk of bias | Low risk of bias | Low risk of bias | Low risk of bias | Serious risk of bias |
| Epstein *et al.* 2022 | Critical risk of bias | Moderate risk of bias | Low risk of bias | Low risk of bias | Low risk of bias | Low risk of bias | Low risk of bias | Critical risk of bias |
| Gimelraikh *et al.* 2022 | Critical risk of bias | Moderate risk of bias | Low risk of bias | Low risk of bias | Low risk of bias | Low risk of bias | Low risk of bias | Critical risk of bias |

## Patient demographics

Two studies observed a statistically significant difference in age between iNormoCa and iHypoCa; patients with iHypoCa were older (Table 1) [35, 36]. However, none of the included studies adjusted for age as a variable in regression modelling. None of the studies demonstrated a statistically significant difference in the Injury Severity Score (ISS) between iHypoCa and iNormoCa patients [35–37]. Across studies, the majority of injuries were from a blunt mechanism (91.9%) with the largest proportion of penetrating injuries being observed in the study by Ciaraglia *et al.* (38/142 (26.7%)) [35].

## Treatments prior to calcium measurement

One study reported the association between prehospital variables and the incidence of iHypoCa [35]. Ciaraglia *et al.* did not observe any significant difference between groups with respect to prehospital (weight-adjusted) crystalloid volume (iHypoCa 8.04 ml/kg (SD 12.6) compared with iNormoCa 9.46 ml/kg (SD 17.91), $p = 0.650$) [35]. The mean interval between injury and hospital arrival was not different between iHypoCa and iNormoCa paediatric patients: iHypoCa– 189.6 minutes (standard deviation (SD) 282.1) compared with iNormoCa– 169.4 minutes (SD 173.9), $p = 0.744$.

## Association with adverse outcomes

The requirement for blood transfusion was reported differently in each study. Ciaraglia *et al.* recorded weight-adjusted transfusion volumes at 4 and 24 hours, and observed at both time

**Table 3. Definitions and incidence of ionised hypocalcaemia in paediatric major trauma patients.**

| Author | Definition of hypocalcaemia | Timing of calcium measurement | Incidence of hypocalcaemia | Incidence of severe hypocalcaemia (<1.00mmol/l) |
|---|---|---|---|---|
| Ciaraglia *et al.* 2023 | iCa <1.00 mmol/l | Within 15 minutes of arrival | 66/142 (46.5%) | 66/142 (46.5%) |
| Epstein *et al.* 2022 | iCa <1.1 mmol/l | Taken in the Emergency Department–exact timing not reported | 24/457 (5.3%) | 3/457 (0.7%) |
| Gimelraikh *et al.* 2022 | iCa <1.16 mmol/l | Taken in the Emergency Department–exact timing not reported | 22/111 (19.8%) | 3/111 (2.7%) |

iCa–ionised calcium concentration

**Table 4. Ionised hypocalcaemia and the association with adverse outcomes in paediatric major trauma patients.**

| Author | Hospital Length of Stay (days) | | | PICU Length of Stay (days) | | | In-Hospital Mortality | | |
|---|---|---|---|---|---|---|---|---|---|
| | iHypoCa | iNormoCa | p | iHypoCa | iNormoCa | p | iHypoCa | iNormoCa | p |
| **Ciaraglia et al. 2023** | Mean 19.75 (SD 20.10) | Mean 15.30 (SD 15.94) | 0.58 | Mean 11.81 (SD 13.29) | Mean 9.11 (SD 10.13) | 0.59 | 21/66 (31.8%) | 19/76 (25.0%) | 0.45 |
| **Epstein et al. 2022** | Mean 14.4 (SD 18.9) | Mean 10.13 (SD 11.61) | 0.43 | Mean 4.5 (SD 12.1) | Mean 3.78 (SD 6.92) | 0.67 | 2/24 (8.3%) | 13/433 (3.0%) | 0.40 |
| **Gimelraikh et al. 2022** | Mean 11 (SD 11) | Mean 10 (SD 12.6) | 0.36 | Mean 4 (SD 4) | Mean 4 (SD 3.75) | 0.43 | 2/22 (9.1%) | 1/89 (1.1%) | 0.18 |

iCa–Ionised calcium; iHypoCa–Low ionised calcium concentration–defined differently by each author; iNormoCa–Normal ionised calcium concentration–defined differently by each author; PICU–Paediatric Intensive Care Unit; SD–standard deviation

intervals that iHypoCa was associated with significantly higher weight-adjusted transfusion volumes ($p = 0.002$ and $p = 0.040$ respectively) [35]. Epstein et al. dichotomously recorded the need for blood transfusion within the ED and found this requirement to be significantly different between iHypoCa (7/24 (29.2%)) and iNormoCa (28/433 (6.5%)) ($p<0.001$) [36]. Gimelraikh et al. recorded the requirement for blood transfusion during the first 48 hours and did not observe this requirement to be significantly different for iHypoCa (6/22 (27.3%)) and iNormoCa (18/89 (20.2%) ($p = 0.67$) [37]. No studies reported whether paediatric patients received exogenous calcium replacement.

For all studies, hospital and PICU length of stay were reported in all studies and were not observed to be different between iHypoCa and iNormoCa patients (Table 4). The length of stay in PICU and the hospital in general, was observed to be greater for the cohort reported by Ciaraglia et al. suggesting a greater burden of injury [35].

## Meta-analysis results for the association of ionised hypocalcaemia with mortality

The $I^2$ value (39%) suggested low heterogeneity. Concerning heterogeneity was also judged unlikely based on non-significant Cochrane's Q ($p = 0.294$), a reassuring Galbraith graph (S3 File), and on the fact that fixed-effect modelling (S3 File) generated similar effect estimates to those calculated using random-effects SJ modelling.

The overall calculation (Fig 2) was judged acceptable given the conclusion that heterogeneity did not preclude generation of a pooled effect estimate for the mortality OR. There was a non-significant ($p = 0.122$) mortality association: iHypoCa mortality OR (with >1 corresponding to higher mortality with iHypoCa) was 2.26 (95%CI 0.80–6.39); full results details are reported in the S3 File. Trim-fill analysis identified possibility of small-study bias, but two imputed studies fell within the region of non-significant $p$ values. With the new study $N$ of five studies the mortality OR 95% CI continued to overlap the null. Full results including contoured funnel plot are presented in the Supplement. Cumulative meta-analysis (S3 File) demonstrated that with increasing study $N$ the effect estimate appeared to stabilize around the pooled effect estimate of approximately 2.2. At all points in the cumulative meta-analysis the effect estimate's 95%CI overlapped zero. Similar findings were calculated when meta-analysis was repeated using omitted-study assessment (S3 File); omission of Ciaraglia et al. resulted in the pooled effect estimate (of the two remaining studies) approaching significance but leave-one-out analysis did not result in any $p$ estimates dropping below 0.05 and all 95%CIs overlapped the null.

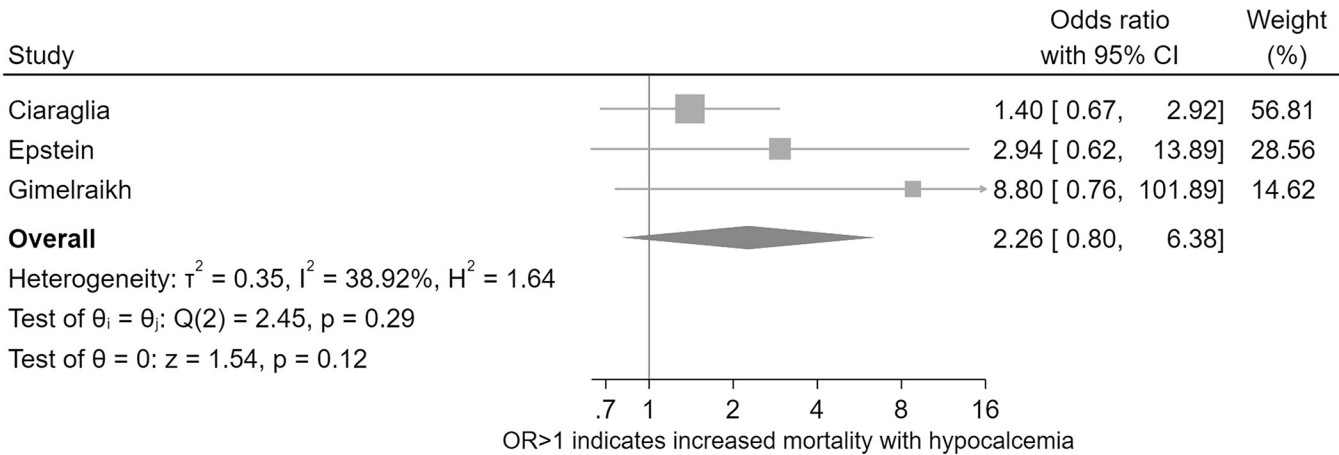

**Fig 2. Forest plot for the endpoint of mortality comparing hypocalcaemic and normocalcaemic paediatric major trauma patients.**

## Association with physiological abnormalities

The presence of haemodynamic instability (an elevated SIPA on arrival) was reported in two studies [35, 36]. Ciaraglia *et al*. observed 28/66 (42.4%) of hypocalcaemic patients to be haemodynamically unstable, and that this was associated with an increased odds of haemodynamic instability when compared to iNormoCa patients OR 3.57 (95% CI 1.65–7.72) [35]. In contrast, Epstein *et al*. observed 6/24 (25.0%) of iHypoCa patients to be haemodynamically unstable and that this was not associated with an increased odds of haemodynamic instability when compared to iNormoCa patients–OR 0.88 (95%CI 0.34–2.27) [36]. Gimelrakih *et al*. did not undertake analysis for haemodynamic instability [37]. No studies reported the requirement for vasoactive medications or the requirement for invasive interventions (e.g., surgical intervention or interventional radiology).

Markers of coagulation were reported in two studies (Table 5) [36, 37]. In these studies, only PTT was recorded in both studies, and neither study observed a significant difference between iHypoCa and iNormoCa paediatric patients.

Ciaraglia *et al*. did not undertake analyses for any markers of clotting, and across studies, PTT was the only clotting test that was recorded in more than one study [35]. Epstein *et al*. measured PTT, INR, and platelets, using INR >1.5 to dichotomise the presence of coagulopathy, observing no significant difference between iHypoCa and iNormoCa paediatric patients [36]. Gimelraikh *et al*. measured PT, PTT and platelets, and also observed no significant difference between the iHypoCa and iNormoCa paediatric patients [37].

## Meta-analysis results for the mean difference in pH between hypocalcaemic and normocalcaemic patients

The $I^2$ value (57%) indicated potential for moderate heterogeneity (Fig 2). Concerning heterogeneity was judged unlikely based on non-significant Cochrane's Q ($p$ = 0.100), a reassuring Galbraith graph (S3 File), and on the fact that fixed-effect modelling (S3 File) generated similar effect estimates to those calculated using random-effects SJ modelling.

The overall calculation (Fig 3) was judged acceptable given the conclusion that marked heterogeneity did not preclude generation of a pooled effect estimate for the mean pH difference. There was a non-significant ($p$ = 0.657) pH difference: pH's standardized mean was calculated

**Table 5. Ionised hypocalcaemia and the association with laboratory abnormalities in paediatric major trauma patients.**

| Author | Cohort size / n | PTT (seconds) | | | pH | | | Lactate (mmol/l) | | |
|---|---|---|---|---|---|---|---|---|---|---|
| | | iHypoCa | iNormoCa | p | iHypoCa | iNormoCa | p | iHypoCa | iNormoCa | p |
| **Ciaraglia et al. 2023** | 142 | Not recorded | Not recorded | N/A | Mean 7.22 (SD 0.17) | Mean 7.28 (SD 0.16) | **0.034** | Mean 5.90 (SD 3.77) | Mean 4.32 (SD 3.11) | **0.009** |
| **Epstein et al. 2022** | 457 | Median PTT 27.7 [IQR 24.5–30.6] | Median 26.1 [IQR 24.1–28.9] | 0.37 | Mean 7.31 (SD 0.14) | Mean 7.31 (SD 0.08) | 0.35 | Mean 3.54 (SD 3.92) | Mean 2.33 (SD 1.71) | 0.23 |
| **Gimelraikh et al. 2022** | 111 | Median 28.3 [26.7–31] | Median 28.6 [IQR 26.3–30.9] | 0.88 | Mean 7.32 (SD 0.09) | Mean 7.3 (SD 0.09) | **0.09** | Mean 2.82 (SD 2.08) | Mean 3.17 (SD 2.53) | 0.67 |

iCa–Ionised calcium; iHypoCa–Low ionised calcium concentration–defined differently by each author; iNormoCa–Normal ionised calcium concentration–defined differently by each author; PTT–Partial Thromboplastin Time; SD–standard deviation

to be 0.08 lower in iHypoCa (95%CI -0.43–0.27). The non-standardized mean difference was calculated as pH 0.01 lower in iHypoCa ((95%CI -0.056–0.035, $p = 0.657$), full results details reported in S3 File).

Trim-fill analysis identified possibility of small-study bias. Two studies were imputed, but with the new study $N$ of five studies the Glass D 95% CI continued to overlap the null. Full results including contoured funnel plot are presented in the S3 File.

Cumulative meta-analysis (details reported in S3 File) demonstrated that with increasing study $N$ the effect estimate approached the null value. At all points in the cumulative meta-analysis the effect estimate's 95%CI overlapped zero. Similar findings were calculated when meta-analysis was repeated using omitted-study assessment (S3 File); omission of any individual study did not result in a finding of statistically significant pH difference in iHypoCa vs iNormoCa.

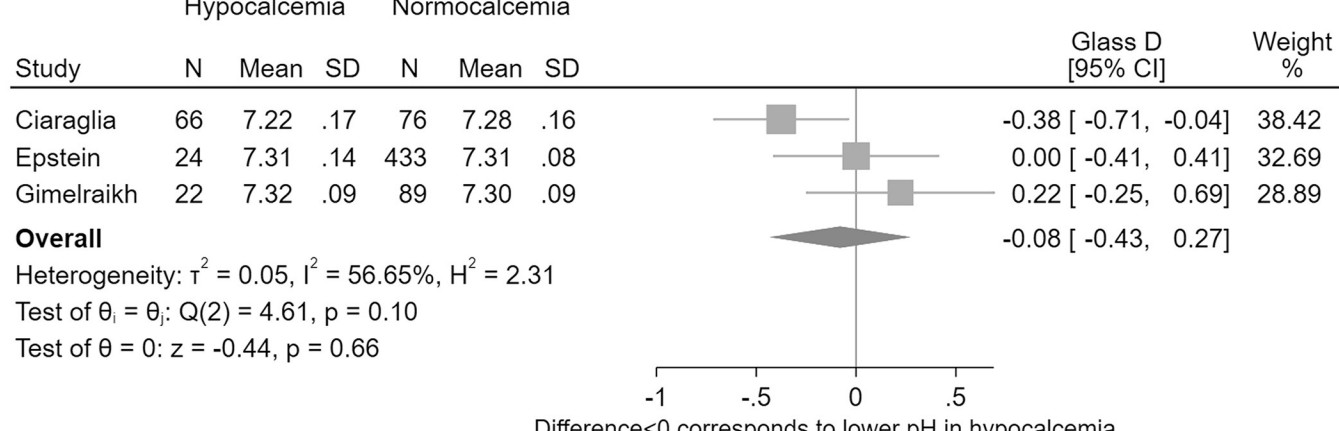

**Fig 3. Forest plot for the endpoint of mean pH difference between hypocalcaemic and normocalcaemic paediatric major trauma patients.**

## Discussion

This review identified three retrospective cohort studies, two studies from Israel and one study from the USA. An iHypoCa was observed in approximately one in six paediatric major trauma patients, and this was reported to be more prevalent in older children. Although there was no significant difference in pH, hospital or PICU LOS, or mortality, there may be an increase in haemodynamic instability and an increased requirement for blood transfusions within the first 24 hours of admission for patients with iHypoCa compared to iNormoCa patients. The included articles do not provide evidence of a causal relationship between iHypoCa and adverse outcomes.

### Incidence of ionised hypocalcaemia

The overall incidence of admission iHypoCa in 112/710 (15.8%) paediatric major trauma patients is lower than that reported in adult major trauma patients (13.0–56.2%) [14–19]. This lower incidence in the paediatric population has also been observed for transfusion-related iHypoCa [34, 38]. Similarly to this review, in a systematic review involving adult patients the definitions and incidence of iHypoCa varied between studies and may reflect a lack of consensus data on what represents clinically significant iHypoCa [17]. Overall, the studies included in the adult systematic review used lower cut-offs for iHypoCa than two out of the three studies included in this review, with one study in this review defining iHypoCa as <1.16mmol/l [37]. The inclusion of these higher cut-off values is important as this reflects a common laboratory reference value and may also inform subsequent work related to mild or moderate hypocalcaemia [39, 40].

In this review two out of three studies observed that older age was associated with iHypoCa [35, 36]. Moreover, the three studies included in this review included patients up to the age of 18 years of age, where physiology is more likely to be similar to adults [35–37]. As such, it remains unclear whether calcium imbalance in paediatric trauma should be considered different to the adult trauma population, and further studies that are powered to detect age-related differences in a younger cohort would be useful to clarify relevance.

The incidence of iHypoCa in each of the studies included in this review varied widely (5.3% to 46.5%) [35–37]. The higher incidence of iHypoCa in the study by Ciaraglia et al. (66/142 (46.5%)) may reflect the exclusion of isolated head injuries, the greater proportion of penetrating injuries in this group (38/142 (26.7%)) and differences in institutional trauma team activation criteria (S4 File) [35]. Penetrating injuries are associated with a higher incidence of iHypoCa [18] and trauma-induced coagulopathy [4]; however, markers of coagulopathy were not reported in this study [32]. Head injuries are recognised to increase the fibrinolytic system and are prone to coagulopathy; the true effect of excluding these isolated injuries in Ciaraglia et al.'s study is therefore unclear [41]. As such, the true incidence of iHypoCa and relevant cut-off values in paediatric major trauma remains unclear; however, with 15.8% of the population being identified as having iHypoCa in the ED, this does not appear to be a rare phenomenon.

### Treatment requirements

Although the association between iHypoCa and the requirement for blood transfusion was explored differently in each study, this was observed to be significant in the first 24 hours, particularly in the first four hours [35, 36]. Among adult major trauma patients multiple studies have observed iHypoCa to be associated with increased transfusion requirements and increased requirement for massive transfusion [13, 17, 42]. This is particularly important when considering early Trauma Induced Coagulopathy [4, 8].

Additional treatments, such as vasoactive medications, or invasive interventions (interventional radiology or surgery) were not recorded in any of the included studies [35–37]. Moreover, none of the studies included in our review explored the effect of exogenous calcium administration. At present, there is not evidence to support calcium supplementation. Moreover, a recent systematic review of empirical calcium administration for adult patients in cardiac arrest has demonstrated harm [43].

## Association with adverse outcomes

This review observed iHypoCa to increase the odds of admission haemodynamic instability two-fold but did not demonstrate any association between hospital or PICU LOS, nor mortality. Calcium concentration play an integral role in both vascular smooth muscle and cardiac contractility [44, 45]. Previous work involving adults has observed both hypotension and a significantly worse shock index amongst hypocalcaemic (iCa <1.0 mmol/l) trauma patients [13, 17–19]. However, these studies used a low cut-off for iHypoCa, with a similar cut-off only used in one out of the three studies explored in this review [33]. Epstein *et al.* used a cut-off of iCa <1.1mmol/l and did not observe a significantly increased difference, however, an overall meta-analysis of both Ciagliaria *et al.* and Epstein *et al.*'s studies did show an overall increased odds of haemodynamic instability [35, 36]. Gimelraikh *et al.* did not undertake analysis for haemodynamic instability and used a cut-off of iCa <1.16 mmol/l, as such, the association between milder iHypoCa and haemodynamic instability is unknown and further work is required to explore this [37].

This review demonstrated a trend towards increased hospital and PICU length of stay but this was not statistically significant. In a large retrospective cohort study involving 1305 injured paediatric patients (median age 11.2 years old, mean ISS 12.14), Cornelius *et al.* observed length of stay to be longer in paediatric patients with iHypoCa (serum calcium <9 mg/dl [2.25 mmol/l] compared to those with iNormoCa [33]. This study also observed an increased requirement for transfusion and for interventional radiology or operative management amongst patients with iHypoCa [33]. However, we excluded this study from this analysis as the authors used total calcium concentration, which may not reflect the physiologically active ionised calcium concentration [11]. Moreover, total calcium concentration have less utility in the ED, with values not being as quickly available as blood gas measurements [10]. The study also included low acuity isolated orthopaedic injuries, so the discrepancy between unwell paediatric patients with iHypoCa may be more pronounced [33]. The lack of difference between groups may reflect the practice of undertaking a conservative and observational approach to paediatric trauma [46, 47].

Although there was a trend towards increased odds of mortality for iHypoCa patients, this review did not observe any significant difference in mortality rates between paediatric patients with iHypoCa and iNormoCa. In contrast, studies in adult trauma patients looking at calcium concentration prior to blood transfusion have demonstrated an association between iHypoCa and mortality [14–16]. Additionally, hypercalcaemia (iCa $\geq$ 1.30 mmol/l) has also been associated with mortality and adverse outcomes in adult trauma patients, suggesting that there is a parabolic association between calcium levels and adverse outcomes [18].

## Association with physiological abnormalities

This review did not demonstrate an association between iHypoCa and coagulopathy, pH, or hyperlactatemia. The presence of coagulopathy was heterogeneously explored between studies, and no studies used thromboelastography (TEG) or rotational thromboelastometry (ROTEM), thus limiting conclusions on the association between iHypoCa and coagulopathy

[35–37]. In contrast, studies involving adult major trauma patients have observed an independent association between iHypoCa and coagulopathy (defined as a PTT $\geq$40 s or an INR $\geq$1.4) on arrival to the ED [17–19].

Acidosis is well-known to impair coagulation, and a positive correlation between iHypoCa and acidosis has previously been demonstrated post-injury [17, 48]. Mechanistically this seems paradoxical, as alkalosis leads to intracellular calcium shifts and iHypoCa, however, acidosis in trauma is largely due to ischaemia and reperfusion injury causing a raised lactate, which chelates calcium and causes pH-dependent calcium binding [6, 13, 17]. There is the potential for both acidosis and alkalosis to confound iCa levels; although a pH calcium correction formula can be used (with around 5% calcium level increase for each 0.1 pH unit decrease), this may risk underestimating iHypoCa and may be less relevant for interpretation in the clinical situation [11, 42]. In the studies included in this review, pH was similar amongst the iHypoCa and iNormoCa paediatric patients, with small standard deviations in individual studies and narrow confidence intervals in the meta-analysis. This suggests that few paediatric patients had severe acidosis. In contrast, acidosis is more prevalent amongst adult major trauma patients and has been associated with iHypoCa in this cohort [16, 18, 19].

It is also notable that none of the studies included in this review reported serum potassium concentrations, with raised concentrations being recognised as deleterious in traumatic coagulopathy it would be beneficial to quantify this in future studies [49].

## Limitations

This review was primarily limited by the small number of available studies, with each study being a single-centre and retrospective design. Moreover, the pooling of data when different definitions of iHypoCa were used limits the true assessment of the incidence and implications of iHypoCa in paediatric trauma. However, the review has the advantage of being the first study to explore this area, and the methodology appropriately reflects the lack of a standardised definition of iHypoCa in current clinical practice.

The inclusion of Ciaraglia *et al's* study which had a minority of patients who received citrated blood products prehospital prior to iCa measurement is potentially a significant confounder [35]. However, the prehospital transfusion volumes were small and similar with no statistically significant difference between the iHypoCa (34/66 (51.5%)) and iNormoCa (36/76 (47.4%)) groups, suggesting this did not impact upon admission iHypoCa rates or outcomes [35]. Moreover, these patients did not receive any exogenous calcium administration prior to calcium measurement. Therefore, the inclusion of these patients in meta-analysis is unlikely to be a significant confounder and is appropriate given the small number of studies available.

Additional limitations relate to the end points for meta-analysis. Given the low overall incidence of mortality, the study is likely underpowered to detect a mortality difference. Moreover, studies only report inpatient mortality, and as such, this may miss later morality in paediatric trauma [50]. Despite this, the study is novel and represents the largest sample size exploring iHypoCa outcomes in the paediatric population. There are also limitations to using pH as a marker of physiological derangement. pH is confounded by unmeasured metabolic and respiratory variables, and pH itself alters iCa levels [11, 51]. However, this was selected due to the availability of data, and has clinical relevance for paediatric trauma given that acidosis may further potentiate trauma-induced coagulopathy [6].

There is also a risk of selection bias, with all studies using trauma team activation as part of the inclusion criteria, since a reasonable percentage of injured paediatric patients are recognised to self-present (with their caregiver) to the ED this is likely to have excluded several seriously injured paediatric patients [52, 53].

## Conclusion

Admission iHypoCa was present in at least one in six paediatric major trauma patients and may be associated with haemodynamic instability and increased blood transfusion requirements. An iHypoCa was not identified to have a statistically significant association with mortality.

## Supporting information

**S1 Checklist. PRISMA checklist.**
(PDF)

**S1 File. Search strategy and results.**
(PDF)

**S2 File. Excel spreadsheet featuring extracted data.**
(XLSX)

**S3 File. Additional statistical analysis.**
(PDF)

**S4 File. Trauma team activation criteria for included studies.**
(PDF)

## Acknowledgments

The authors would like to acknowledge and thank Catherine Hancox (Academic Librarian) and the Defence Medical Services Library Team for their assistance with the search strategy. Dr Peter Brooke is acknowledged for his review of the draft of this review, and Dr Mark Lyttle for his review of the protocol. The authors would also like to acknowledge and thank the authors of the included studies, in particular Dr Angelo Ciaraglia, Dr Danny Epstein, and Dr Nir Samuel, for their support and provision of data.

PROSPERO registration number: CRD42023425172

Protocol published in BMJ Open (doi:10.1136/bmjopen-2023-077429)

## Author Contributions

**Conceptualization:** Owen Hibberd, Edward B. G. Barnard.

**Data curation:** Owen Hibberd, James Price, Stephen H. Thomas, Tim Harris, Edward B. G. Barnard.

**Formal analysis:** Owen Hibberd, James Price, Stephen H. Thomas.

**Investigation:** Owen Hibberd, James Price, Stephen H. Thomas, Tim Harris, Edward B. G. Barnard.

**Methodology:** Owen Hibberd, James Price, Stephen H. Thomas, Tim Harris, Edward B. G. Barnard.

**Project administration:** Owen Hibberd, James Price, Edward B. G. Barnard.

**Resources:** Stephen H. Thomas.

**Software:** Stephen H. Thomas.

**Supervision:** Stephen H. Thomas, Tim Harris, Edward B. G. Barnard.

**Visualization:** Owen Hibberd, Edward B. G. Barnard.

**Writing – original draft:** Owen Hibberd, James Price, Tim Harris, Edward B. G. Barnard.

**Writing – review & editing:** Owen Hibberd, James Price, Stephen H. Thomas, Tim Harris, Edward B. G. Barnard.

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
