## [Decision Letter · Decision Letter 0]

9 Apr 2024

PONE-D-24-02104The Incidence of Admission Ionised Hypocalcaemia in Paediatric Major Trauma – a Systematic Review and Meta-AnalysisPLOS ONE

Dear Dr. Hibberd,

Thank you for submitting your manuscript to PLOS ONE. After careful consideration, we feel that it has merit but does not fully meet PLOS ONE’s publication criteria as it currently stands. Therefore, we invite you to submit a revised version of the manuscript that addresses the points raised during the review process.

**Thanks for your interesting manuscript. Few minor comments.**

We look forward to receiving your revised manuscript.

Kind regards,

Jean Baptiste Lascarrou

Academic Editor

PLOS ONE

Journal Requirements:

Reviewers' comments:

Reviewer's Responses to Questions

**Comments to the Author**

1. Is the manuscript technically sound, and do the data support the conclusions?

Reviewer #1: Yes

Reviewer #2: Yes

2. Has the statistical analysis been performed appropriately and rigorously? 

Reviewer #1: Yes

Reviewer #2: Yes

3. Have the authors made all data underlying the findings in their manuscript fully available?

Reviewer #1: Yes

Reviewer #2: Yes

4. Is the manuscript presented in an intelligible fashion and written in standard English?

Reviewer #1: Yes

Reviewer #2: Yes

5. Review Comments to the Author

**Reviewer #1:** - It looks like you have miscalculated the incidence of iHypoCa. Epstein's article reports 21 patients with mild iHypoCa (1.0-1.1mmol/L) and 3 patients with severe iHypoCa (<1.0mmol/L). This makes a total of 24 patients with iHypoCa <1.1mmol/L and not 21 as you have stated in your paper. On line 325 you correctly refer to the total number of 24.

This affects the overall incidence of iHypoCa. It probably doesn't change the conclusion, but this error should be corrected.

- All 3 included articles report on the incidence of severe iHypoCa <1.0mmol/L. I would recommend that you also include the incidence with this identical definition in your paper.

- To support your statement that children may be more sensitive to iHypoCa (line 107), it is suggested to include another reference in addition to Barcelona et al. This is because Barcelona et al. only mention neonates as being more sensitive to citrate-induced iHypoCa, which is a very limited group in the overall paediatric trauma population.

Furthermore, Hobbs et al reported that transfusion-related iHypoCa was less prevalent in children than in adults. You also mention an increased incidence of iHypoCa with older age.

Therefore, it remains unclear whether calcium imbalances are truly relevant for the paediatric population and whether they should be studied differently from the adult trauma population.

- In the first paragraph of the introduction (line 93), it is mentioned that treatment of hypocalcemia is recommended to deliver effective trauma resuscitation. However, there is currently no evidence to support a causal relationship between hypocalcemia and worse outcomes, which would justify calcium supplementation. It is important to note that association with worse outcomes does not necessarily mean that hypocalcemia is the cause. Additionally, a similar relationship with iHyperCa was found in the adult population.

Similar, in this review in the paediatric population, only an association between iHypoCa and haemodynamic instability and increased blood transfusion was found. The included articles do not provide any evidence of a causal relationship. It is recommended to comment on this in your article.

- The difference in the incidence of iHypoCa between Chiaraglia's study and the other two is remarkable, particularly considering the lower ISS in Chiaraglia's study. Although the increased incidence of penetrating trauma and different trauma criteria may have contributed, it is worth considering whether other factors may have played a role, like the exclusion of iTBI. Epstein's study found that iTBI accounted for 17% of his trauma population. Can you comment on this?

- Line 359 should start with a capital letter.

- In the paper, both 'hypocalcemia' and 'iHypoCa' are used. It is recommended to consistently use 'iHypoCa'.

**Reviewer #2:** This is a Systematic review and Meta-Analysis of ionised hypocalcaemia in pediatric trauma. The authors report that ionised hypocalcaemia was present in one in six Paediatric major trauma and that statistically there was no significant correlation with mortality or difference with pH.

The SR is in firm compliance with PRISMA guideline in this SR, and the included papers are composed of quality-assured articles with no predatory journal. The Clinical Question presented in the paper is also clearly and soundly answered with research-derived answers and references to the limitaiton that should be included in the paper.

There were several points that I would like to see mentioned in the peer review process, and I will discuss them below.

(1) You included a study in which the cutoff value for iCa was 1.16 mmol/L, which I think is rather high. I think that 1mmol/L should be the standard for adults as well.

(2) The inclusion criterion of trauma team activation is ambiguous. Shouldn't it be detailed?

(3) You mentioned pH in the Discussion, but didn't you consider correcting the iCa value for pH in the first place?

(4） Shouldn't there be a slight mention of the Sidik-Jonkman approach? (This is not a common approach.)

(5) You incorporated only one study in which a single head injury was excluded. Head trauma increases the fibrinolytic system, which is prone to coagulopathy and, of course, affects iCa levels in the adult population. What do you think about the possibility of underestimating the results?

(6) With regard to pH, we have to say that statistical heterogeneity is SUBSTANTIAL heterogenenity.

What do you think about the possibility of pH direct confounding for iCa?

I would appreciate if you could explain or add to these points.

6. PLOS authors have the option to publish the peer review history of their article (what does this mean?). If published, this will include your full peer review and any attached files.

Reviewer #1: No

Reviewer #2: **Yes: **Toshiro Imamoto

---

## [Author Response · Author response to Decision Letter 0]

12 Apr 2024

Thank you. Detailed response to reviewers is attached.

---

## [Decision Letter · Decision Letter 1]

19 Apr 2024

The Incidence of Admission Ionised Hypocalcaemia in Paediatric Major Trauma – a Systematic Review and Meta-Analysis

PONE-D-24-02104R1

Dear Dr. Hibberd,

We’re pleased to inform you that your manuscript has been judged scientifically suitable for publication and will be formally accepted for publication once it meets all outstanding technical requirements.

Kind regards,

Jean Baptiste Lascarrou

Academic Editor

PLOS ONE

Additional Editor Comments (optional):

Reviewers' comments:

Reviewer's Responses to Questions

**Comments to the Author**

1. If the authors have adequately addressed your comments raised in a previous round of review and you feel that this manuscript is now acceptable for publication, you may indicate that here to bypass the “Comments to the Author” section, enter your conflict of interest statement in the “Confidential to Editor” section, and submit your "Accept" recommendation.

Reviewer #1: All comments have been addressed

Reviewer #2: All comments have been addressed

2. Is the manuscript technically sound, and do the data support the conclusions?

Reviewer #1: Yes

Reviewer #2: Yes

3. Has the statistical analysis been performed appropriately and rigorously? 

Reviewer #1: Yes

Reviewer #2: Yes

4. Have the authors made all data underlying the findings in their manuscript fully available?

Reviewer #1: Yes

Reviewer #2: Yes

5. Is the manuscript presented in an intelligible fashion and written in standard English?

Reviewer #1: Yes

Reviewer #2: Yes

6. Review Comments to the Author

Reviewer #1: Congratulations on your excellent article.

All comments have been well addressed, and the changes have further enhanced its quality.

Reviewer #2: Thanks for your hard work Revise. I think this is a very good paper, and I appreciate the thoughtful answers to the questions I asked and suggestions I made.

I think the conclusions are also scientifically valid arguments without being over statment. It is common in other fields of trauma for adults and children to have different views on the importance of calcium in severe trauma, and I hope that you have completed a systematic review and meta-analysis of the current position of calcium in pediatric trauma, and that this paper will be a landmark for future research on calcium in pediatric trauma. We believe that this paper will be a landmark in the future of calcium research in pediatric trauma.

7. PLOS authors have the option to publish the peer review history of their article (what does this mean?). If published, this will include your full peer review and any attached files.

Reviewer #1: No

Reviewer #2: **Yes: **Toshiro Imamoto
